



# On the mesoscale monitoring capability of Argo floats in the Mediterranean Sea

Antonio Sánchez-Román[1], Simón Ruiz[1], Ananda Pascual[1], Baptiste Mourre[2] and Stéphanie Guinehut[3]

[1] IMEDEA (UIB-CSIC), Mallorca, Spain
[2] SOCIB, Mallorca, Spain
[3] CLS, Toulouse, France

*Correspondence to*: Antonio Sánchez-Román (asanchez@imedea.uib-csic.es)

## Abstract

In this work an Observing System Simulation Experiment (OSSE) approach is used to investigate the Argo array spatial sampling necessary in the Mediterranean Sea to recover the mesoscale signal as seen by altimetry. The monitoring of the mesoscale features is not an initial objective of the Argo network. However, it is an interesting question in the perspective of future network extensions in order to improve the ocean state estimates. A quality assessment of the performances of the altimeter product is carried out to quantify the differences between Argo and altimetry needed to conduct the simulation experiments. The method used here to evaluate the altimeter data is based on the comparison of Sea Level Anomalies (SLA) from altimetry and Dynamic Height Anomalies (DHA) referred to both 400 and 900 dbar computed from the in-situ Argo network. A standard deviation of the differences between SLA and DHA of 4.92 cm is obtained when comparing altimetry and Argo data referred to 400 dbar. The simulation experiments show that a configuration similar to the current Argo array in the Mediterranean (with a spatial resolution of 2° × 2°) is only able to recover the large-scale signals of the basin. On the contrary, the SLA field reconstructed from a 0.75° x 0.75° Argo network can retrieve most of the mesoscale signal. Such an Argo array of around 450 floats in the Mediterranean Sea would be enough to recover the SLA field with an RMSE of 3 cm for spatial scales higher than 150 km, similar to those captured by the altimetry.

Keywords: Mediterranean Sea, Observing System Simulation Experiment, altimetry errors, in-situ measurements, profiling float, Array design.





## 1. Introduction

In the decade of 1990s, the Topex/Poseidon (T/P) mission combined with the ERS-1/2 satellites changed our sight of the sea level and ocean circulation fluctuations radically. Since then, altimeter missions have furnished accurate measurements of Sea Surface Height (SSH). Mean Sea Level (MSL) has been monitored form multiple datasets provided by different altimetry missions presently in flight (Jason 2 and 3, HY-2A, CryoSat-2, Sentinel-3A, SARAL/Altika) and those no longer supplying data (the aforementioned T/P, ERS-1 and 2, Jason 1, Envisat, Geosat Follow-On) [*Legeais et al.,* 2016]. Altimetry resolves the mesoscale thanks to a finest spatio-temporal sampling. Nevertheless, even though SSH estimates are becoming more precise, the uncertainty associated with altimeter measurements and the geophysical altimeter corrections applied in the SSH computation remains relatively high [*Ablain et al.,* 2009; *Couhert et al.,* 2014; *Legeais et al.,* 2014; *Rudenko et al.,* 2014]. For this reason, some external and independent measurements provided by in-situ observations and numerical models are required to calibrate and validate the altimeter Sea Level Anomaly (SLA) data. These comparisons allow us to obtain the altimetry errors relative to the external measurements and provide an improved picture of SSH that can be used for global and regional studies.

Tide gauges data are usually considered [e.g. *Mitchum* 1998, 2000; *Nerem et al.,* 2010; *Arnault et al.,* 2011; *Villadeau et al.,* 2012] because they furnish high temporal resolution time series of Sea Surface Height (SSH) in coastal areas. However, these instruments are not homogenously allocated over the coasts. A complementary approach can be done by using (i) in-situ Dynamic Heights Anomalies (DHAs) derived from the Temperature and Salinity (T/S) vertical profiles computed from the Argo network [see e.g. *Guinehut et al.,* 2012; *Valladeau et al.,* 2012; *Legeais et al.,* 2016] and glider measurements [e.g. *Ruiz et al.,* 2009a, 2009b; *Bouffard et al.,* 2010] or (ii) velocity data provided by drifters [e.g. *Escudier et al.,* 2013; *Troupin et al.,* 2015].

In this study, we will focus on the altimeter products for the Mediterranean basin. The Mediterranean Sea is a semi-enclosed basin connected with the Atlantic Ocean through the Strait of Gibraltar. It also communicates with the Black Sea through the Turkish Bosphorus and Dardanelles Straits. The Sicily channel separates the eastern and western basins [*Criado-Aldeanueva et al.,* 2012]. The basin-scale circulation of the Mediterranean interacts with sub-basin scale and mesoscale processes, then forming a highly variable general circulation. As a consequence, the Mediterranean Sea is a particularly interesting area for among others



physical studies since most of the ocean processes that occur in the world ocean can be also
found in this basin. Therefore, the Mediterranean can be considered as a reduced scale ocean
laboratory, where processes can be characterized with smaller scales than in other ocean
regions [*Malanote–Rizoli et al.*, 2014]. In this context, the internal Rossby Radius of
deformation in the basin is *O(10–15 km)*, this being four times smaller than typical values for
much of the world ocean according to *Robinson et al.* [2001]. This fact promotes that in the
Mediterranean Sea the spatial resolution of the lagrangian profiling floats of the Argo
programme, which consists of a global network of more than 3000 operating floats [*Roemmich*
*et al.*, 2009; *Riser et al.*, 2016] drifting with less than 3 degrees mean spacing, should be
reduced four times compared to the open ocean. The Argo programme is a major component
of the Global Ocean Observing System and aims to monitor the changing temperature and
salinity fields in the upper part of the ocean [*Riser et al.*, 2016]. The majority of the profiling
floats used in Argo are programmed to drift at a nominal depth (known as the parking depth)
of 1000 m [*Riser et al.*, 2016].  They collect temperature and salinity data every 10 days from
the upper 2000 m of the world oceans in order to observe the slow evolution of the large-scale
ocean structure.
Argo and satellite altimetry are entirely complementary. The combination of in-situ Argo
data with SSH anomalies derived from satellites allows us to construct time series of the
dynamical state of the ocean circulation [*Riser et al.*, 2016]. At present, Argo data are
systematically used together with altimeter data to describe and forecast the 3D ocean state,
for ocean and climate research and for sea level rise studies [see e.g. *Guinehut et al.*, 2012; *Le*
*Traon*, 2013]. Although Argo does not resolve the mesoscale, the joint use of Argo, altimetry
and numerical models, through effective data assimilation techniques can provide a good
representation of mesoscale temperature and salinity fields. This fact demonstrates the very
strong and unique complementarities of the two observing systems [*Le Traon*, 2013].
The Argo network in the Mediterranean Sea consists presently of around 80 operating
floats       deployed       in       the       frame       of       the       MedArgo       program
(http://nettuno.ogs.trieste.it/sire/medargo/active/index.php).   The   specific   semi-enclosed
morphology with a large fraction of coastal areas, shallow bathymetry and circulation
structures of the basin make profilers programmed with the Argo standard global parking
depth of 1000 m not appropriate for this program [*Poulain et al.*, 2007]. This is why a parking
depth of 350 m was chosen for the Mediterranean basin. The objective was to track the
intermediate waters throughout the Mediterranean which are mostly composed by Levantine
Intermediate Water (LIW). This water mass is formed during winter convection in the northern



Levantine sub-basin being a crucial component of the Mediterranean thermohaline "conveyor
belt" circulation [*Poulain et al.*, 2007]. According to the small radius of deformation of the
Mediterranean compared with the open ocean, the current number of operating floats in the
basin (equivalent to an average spatial resolution of around 2 degrees) improves the global
coverage of the Argo network. Nonetheless, it is not enough to properly capture the significant
mesoscale circulation features of the basin.

7        The aim of this paper is to investigate which Argo design sampling in the Mediterranean

Sea is necessary to recover the mesoscale signal as seen by altimetry. The monitoring of the
mesoscale structures is not an initial target of the Argo network [*Riser et al.*, 2016]. However,
this is an interesting question in the perspective of future network extensions in order to
improve ocean state estimates. Actually, the Argo Steering Team has recently provided a
roadmap for how the Argo mission might expand in the near future [*Riser et al.*, 2016].
According to these authors, one of the proposed projects is to support an increase in the
spatial sampling resolution in particular areas of the word ocean. The objective is the
improvement of our view of the complex structure of oceanic variability at spatial scales lesser
than the climate scale.

17       To accomplish the proposed aim, we conduct several Observing System Simulation

Experiments (OSSEs) in the basin. OSSEs provide a methodology to evaluate and design
optimum sampling strategies in ocean observing systems (OOS) [*Alvarez and Mourre*, 2012].
Usually, the method consists in considering the outputs of an ocean model simulation of the
area monitored by the OOS as ''truth.'' Virtual observations from different ocean observing
platforms in the OOS are then simulated from the model run and analysed in the same manner
than real data [e.g. *Alvarez and Mourre*, 2012]. OSSEs have been used in oceanography to
analyse the impact of different components of the global OOS for ocean analysis and
forecasting (see e.g. *Oke and Schiller* [2007]; *Guinehut et al*. [2012]; *Alvarez and Mourre*
[2012]; *Ninove et al*. [2015]; *Oke et al.*, [2015a] or *Oke et al.*, [2015b]). Here, however, a
different approach will be followed: we simulate the Argo observing system in the
Mediterranean based on specific altimetry gridded merged product for the Mediterranean Sea
and not from an ocean model simulation. This approach is similar to the one followed by
*Pascual et al.*, [2009]. These authors evaluated the quality of global real-time altimetric
products by comparing them with independent in-situ tide gauges and drifter data. Our study
will then assess the scales covered by altimetry which are larger than 100 km [*Pujol and*
*Larnicol*, 2005]. Notice that the scales mentioned in this paper allude to a definition based on
the diameter of individual structures, usually referred to as "feature scales".



The paper is organized as follows: the datasets are described in Section 2. Section 3 details
both the processing sequence developed to compare the altimeter data with Argo in-situ
measurements and the quantification of the differences between $Argo - SLA$. These
differences are needed to conduct the OSSEs. Thus, a quality assessment of the performances
of the altimeter product in the Mediterranean Sea is performed in the first part of this study.
The method used here to evaluate the altimeter data is based on the comparison of SLAs from
altimetry and DHAs computed from the in-situ Argo network. Section 4 is devoted to the
experiments conducted to recover the SLA fields in the basin from the different configurations
of the simulated Argo arrays. Finally, discussion and suggestions to the Argo community
regarding future prospects of the in-situ network in the Mediterranean Sea are given in Section

11   5.

## 2.  Datasets

### 2.1 ARGO dataset

We use delayed mode quality-controlled T/S profiles from 2003 to middle 2015 as
obtained from the Coriolis Global Data Assembly Centre (www.coriolis.eu.org, ARGO GDAC
global distribution database) in the Mediterranean Sea (Figure 1). Dynamic Height (DH) was
computed at 5 m depth as an integration of the pressure, temperature and salinity vertical
profiles through the water column using a reference level at 400 dbar and 900 dbar (close to
400m and 900m, respectively). The choice of these reference levels is conditioned by the
availability of the climatology used to compute DH anomalies. This issue will be addressed
later. An additional quality control criterion relative to both the profile's position and the T/S
measurements was applied: only profiles with a quality position flag of 1 (good data) were
employed. The major restriction, however, comes from the salinity data close to the sea
surface. Profiles exhibiting salinity flags of 3 or 4 (bad data) in the first 5 meters of the water
column were removed before the DH computation. As a result of this additional quality check,
194 Argo floats and about 17000 T/S profiles distributed over almost the whole Mediterranean
basin are available to compute DH. Their deployment's temporal evolution is shown in Figure
2. More than 90 floats and almost 9000 profiles have been deployed in the last three years of
the period investigated. They represent more than 50 % of the Mediterranean Argo network.
Actually, the number of both floats and profiles has been systematically increasing from 2008
until 2015 reaching its maximum value in 2014 (36 floats deployed and nearly 4000 profiles
carried out).



To calculate a consistent DHA with the altimeter SLAs, we use a mean dynamic height as a
reference computed through a synthetic climatology approach [*Guinehut et al*., 2006]. The
method to compute the synthetic climatology described in *Guinehut et al*. [2006] consists in
the combination of altimeter SLA with simultaneous in-situ dynamic height in order to
compute a mean dynamic height, which is referred to the time period spanning from January
2003 to December 2011. This climatology presents a global coverage and it has been recently
used by *Legeais et al*. [2016] to analyse global altimetry errors by using Argo and GRACE data.
In this paper we will test the mean dynamic height computed in the Mediterranean Sea at 400
dbar and 900 dbar to estimate DHAs.
**2.2 Altimeter measurements**
Radar altimeters provide SSH measurements that are not directly comparable with in-situ
measurements. Therefore, they must be first referenced and corrected from geophysical
signals in order to determine SLAs. In this work, we use SLAs obtained from SSALTO/DUACS
multimission (Saral, Cryosat-2, Jason-1, Jason-2, T/P, Envisat, GFO, ERS-1, ERS-2, and Geosat)
specific reprocessed gridded merged product (level 4) for the Mediterranean Sea. This product
is available in the Mean Sea Level Anomaly (MSLA) section of the Archiving, Validation and
Interpretation of Satellite Oceanographic website (AVISO, http://www.aviso.altimetry.fr). It
has been computed with respect to a twenty-year mean referred to the period 1993 – 2012. A
comprehensive description of SSALTO/DUACS is given in *Pujol et al*. [2013] and *Pujol et al*.
[2016]. The spatial resolution of the dataset is ⅛° × ⅛° and the time period used in this work
spans from January 1993 to April 2014. The quality of this product can be estimated among
others by comparison with in-situ Argo data. To perform this comparison, it is critical that both
types of data have the same interannual temporal reference [*Legeais et al*., 2016]. Thus, the
temporal reference of the altimeter SLA is adapted to the time period spanning from 2003 to
2011 (reference period of the synthetic mean Argo dynamic height). To do that, we subtract
the mean of altimetric SSALTO/DUACS maps (mean value of 3.54 cm for the whole basin) over
2003 – 2011 from the original SLA time series [*Valladeau et al*., 2012].  On the other hand, the
physical content captured by altimetry and Argo profiles is not precisely the same [*Dhomps et*
*al*., 2011] because the barotropic and the deep steric (deeper than the reference level of the
Argo DHA) contributions are missing from the Argo measurements. Therefore, the comparison
of altimeter SLA and in situ Argo DHA is used to detect relative anomalies in altimeter data and
not absolute bias [*Valladeau et al*., 2012].  This comparison allows us to obtain both the
instrument and the representation errors which are needed to perform the OSSEs.
Representation error can be defined as the component of observation error due to unresolved





scales and processes [*Oke and Sakov*, 2008]. In other words, it is the part of the true signal that
cannot be represented on the chosen grid due to limited spatial and temporal resolution.

## 3. Error estimates from comparison of Argo dynamic heights and altimetry sea level anomalies

This section focuses on the comparison of altimetry data with Argo dynamic height in
order to estimate the errors in Argo – SLA differences needed to conduct the OSSEs. In
addition, this analysis can contribute to validate satellite SLAs with an increased confidence. A
sensitivity analysis of the method of comparison of both datasets is provided. This analysis
mainly focuses on the impact of the reference depth selected in the computation of the Argo
DH on the comparison with specific altimetric SLA gridded merged product for the
Mediterranean Sea.

### 3.1   Method for comparing Altimetry and in-situ Argo data

The comparison method of altimetry with Argo data consists in co-locating both types of
datasets since spatial and temporal sampling of altimetry and Argo data are different
[*Valladeau et al.*, 2012]. Altimeter grids and synthetic climatologies were spatially and
temporally interpolated at the position and time of each in situ Argo profile, which is
considered as reference, by using a mapping method based on an optimal interpolation
scheme. This considerable reduces errors due to different sampling characteristics of altimeter
and in-situ data. The period investigated spans from January 2003 (beginning of the Argo
dataset) to April 2014 (ending of the altimetric data used in this study). Then, statistics
analyses are performed between both datasets. Co-located altimeter and Argo DH differences
are analysed in terms of the standard deviation (STD) for the two reference levels used to
compute DHAs from the Argo profiles (namely 400 and 900 dbar). In addition, the robustness
of the results was investigated by computing means of a bootstrap method with $10^3$ random
samples taken from the original SLA-DHA series (see details of the method in *Efron and*
*Tibshirani* [1993]). The studies conducted include: (i) the assessment of the method of
comparison between Altimetry and Argo data in the Mediterranean Sea; and (ii) the evaluation
of the impact of the reference depth selected in the computation of the Argo dynamic height.

### 3.2   Sensitivity to the reference depth for the integration of the Argo dynamic height

The integration of the Argo T/S profiles for the computation of the in-situ dynamic heights
requires a reference level (pressure) where null horizontal velocities are assumed [*Legeais et*



*al*., 2016]. As a rule, the deeper the reference level, the more information from the T/S profiles
is considered. This involves a well sampled steric signal through the water column. However, a
lower number of vertical profiles (those that reach the reference level) are used in the
computation. On the contrary, shallower reference levels allow us to use more floats, although
the vertical steric signal will be less sampled. Thus, we aim at determining the impacts of a
given reference depth of integration on the Argo spatial sampling and on the comparison with
altimeter data in the Mediterranean basin.

8       As it was aforementioned, the choice of a deep reference level for Argo DHAs provides a

better estimation of the baroclinic signal. This is more in agreement with the observed signal
by altimetry [*Legeais et al*., 2016]. Therefore, we conduct the analysis on DH comparison
computed from Argo data referred to the deeper available reference depth of 900 dbar (nearly
900 m) and the specific altimetry product for the Mediterranean Sea. Results are reported in
Table 1. The number of T/S Argo profiles used to compute DH (those that reach at least 900 m
depth) was 416, corresponding to 23 floats. The standard deviation of the differences between
DH from altimetry and Argo (SLA minus DHA) for the common period investigated (from
January 2003 to April 2014) was 5.31 cm. It is equivalent to more than 95 % of SLA signal
variance. The correlation between both datasets was 0.80.
In order to study the impact of the reference level, we repeated the analysis using the
shallower reference level of 400 dbar (almost 400 m) for the Argo anomalies but using the
same array of Argo profiles reaching 900 m. Now, 24 floats and 479 profiles are available to
compare with altimetry due to the synthetic climatology used to compute DHA referred to 900
dbar (see Table 1). Nonetheless, we kept the same number of floats and profiles than in the
previous computation in order to make both results comparable. The standard deviation of the
differences between SLA and DHA referred to 400 dbar computed from profiles spanning until
900 m depth was 5.04 cm (see Table 1). It represents an improvement of nearly 10 % in terms
of signal variance with respect to the STD diff. computed from Argo DHA referred to 900 dbar
(5.31 cm). Moreover, the correlation coefficient increased from 0.80 to 0.82. These slightly
better results (also confirmed from the bootstrap analyses) show that in the Mediterranean
basin, it will be advisable to compare SLA from altimetry with DHA from in-situ Argo data
referred to 400 dbar.
Consequently, DHA referred to 400 dbar was recomputed but using all the available
profiles reaching 400 m depth. Now, the number of T/S Argo profiles used to compute DH
increased to 2258, thus corresponding to 41 Argo floats. Notice that this more comprehensive




number of Argo profiles is almost 6 times larger than the profiles used to compute DHAs
referred to 900 dbar. The standard deviation of the differences of SLA– DHA was 4.92 cm
while the correlation between both datasets decreased to 0.76. We will consider this STD
value as the mean error of the Argo – SLA differences in the Mediterranean and therefore it
will be used to perform the OSSEs. Notice that this result represents an improvement of 14 %
in terms of signal variance with respect to the one obtained from the differences between SLA
and DHA referred to 900 dbar. This is an unexpected result since the larger thickness of the
water column integrated in the latter should promote a lower value of STD. A plausible
explanation of this outcome will be done in Section 5.
## 4- Impact of the number of Argo floats on the reconstructed SLA fields
In this section we aim to investigate which configuration in terms of spatial sampling of the
Argo array in the Mediterranean Sea will properly reproduce the mesoscale dynamics in this
basin, which is comprehensively captured by new standards of specific altimeter products for
this region. To do that, several OSSEs have been conducted to simulate the Argo observing
system in the Mediterranean assuming altimetry data computed from specific reprocessed
gridded merged product for the basin as the "true" field.
### 4.1 Experiments design
In a first step, OSSEs have been performed for daily SLA maps along 2014 by applying the
Optimal Interpolation (OI) technique. The region considered covers the entire Mediterranean
basin. The original altimetry dataset has a spatial resolution of ⅛° × ⅛° and presents 17283 grid
points (see Table 2). Daily SLA maps were sub-sampled with the different spatial resolutions
displayed in Table 2 in order to reproduce some possible configurations of the Argo array
network in the Mediterranean. The stations (grid points) associated with each sub-sampled
field (figures not shown) will simulate the positions of the Argo floats over a regular grid.
Before the computation, the sub-sampled daily SLA maps were perturbed with a random
noise by using a normal distribution function only depending on the standard deviation of the
differences of SLA – DHA (4.92 cm) computed in Section 3. This STD diff. corresponds to the
sum of the instrument and the representation errors. Then, seven experiments were
conducted to reconstruct the different sub-sampled daily SLA fields in the Mediterranean
along 2014 with a spatial resolution of ⅛° × ⅛° by applying the OI technique. The parameters
used for the computation of the reconstructed fields were the following: (i) the first guess used
to obtain the statistically null-mean residuals was computed by fitting a polynomial of degree



1. This first guess will be subsequently added after the computation to recover the total daily
field; (ii) the filtering scale was set to be twice over the spatial distance between stations
(according to the box size used in each experiment). Table 2 summarizes the filtering scale
used to compute the recovered SLA fields in the different reconstructions; (iii) the spatial scale
of correlation between stations was determined from a Gaussian correlation curve computed
as follows:

$$W = e^{-d^2/2 \cdot S^2} \tag{1}$$

where $d$ is the mean distance between stations and $S$ the spatial scale of correlation. In order
to determine the more suitable spatial scale of correlation for the Mediterranean basin we
computed the correlation curve $W$ for spatial scales varying from 15 km to 50 km. The mean
distance between stations ranged between 0 km and 100 km. Then we compared these
correlation curves with the one obtained for altimetric data computed for the same distances
between stations as follows:

$$COR(x) = \left[1 + ar + \tfrac{1}{6}(ar)^2 - \tfrac{1}{6}(ar)^3\right]e^{-ar} \tag{2}$$

with $r = x/L$ and $a = 3.337$ ; where $x$ is the spatial coordinate of the studied point, and L is
the zonal correlation scale (km) of the Mediterranean basin (100 km). The reader is referred to
*Pujol and Larnicol* [2005] for a more detailed description of this computation. Figure 3 shows
the correlation curve computed for the altimetric data from Eq. (2) and the best fitting curve
obtained from Eq. (1), which corresponds to a spatial correlation scale of 40 km. Therefore, the
*S* parameter was set to 40 km in all the experiments. (iv) the last parameter to include in the
experiments is the noise to signal variance ratio (γ), defined as the ratio between the Argo
error and the altimetry variance. The former can be established as the variance of the
differences between SLA and DHA in the Mediterranean. This parameter is estimated from the
standard deviation of SLA-DHA differences (4.92 cm) computed in Section 3. As a result, we
obtain γ=0.9 as the true value for the datasets used here (see further details about this
parameter in *Gomis et al.* [2001]).
Finally, the retrieved daily SLA maps for 2014 were compared with the original ones (also
interpolated to a spatial resolution of ⅓° × ⅓°) in order to compute    the root mean square
errors (RMSE) associated with the recovered maps from the sub-sampled fields. This
procedure will let us establish the spatial resolution that better captures the mesoscale
dynamics in the Mediterranean with a feasible number of stations simulating the locations of
Argo floats.





**4.2 Impact of the grid box size on analysed SLA fields**
In this section we will discuss the impact of the spatial resolution of the sub-sampled SLA
fields on the retrieval of mesoscale signals in the Mediterranean basin. As a previous step, the
RMSE obtained for the seven experiments will be analysed. The 2014 yearly mean values of
the RMSE associated with the altimetry maps recovered from the different sub-sampled fields
and their annual variability are displayed in Figure 4. Maximum mean RMSE larger than 4 cm
(equivalent to 79 % of SLA signal variance) are obtained for the maps recovered from the sub-
sampled field reproducing the current spatial resolution of the Argo array in the
Mediterranean (2° × 2°). Therefore, this spatial configuration only retrieves 21 % of SLA signal
variance due to a poorer capture of the mesoscale features. These maps also exhibit the larger
annual variability. This is an expected result that can be explained by both the challenge of
reconstructing the same scale signals with only 69 stations (grid points) and the larger filtering
scale (around 450 km) used in the experiment (see Table 2). The mean RMSE of the recovered
maps exponentially decays as the box-size of the sub-sampled altimetry fields diminishes and
therefore, the number of stations enhances. As a result, the mean RMSE reaches an
asymptotic value of 2.4 cm (equivalent to 28.7 % of SLA signal variance) for the SLA maps
retrieved from the sub-sampled fields with a box-size of 0.4° × 0.4°. This configuration is
equivalent to 1458 stations and captures 71.3 % of SLA signal variance. The standard deviation
of the RMSE follows the same pattern exhibiting a minimum annual variability for this spatial
resolution.
Figure 5 shows an example of the altimetry maps recovered from the sub-sampled SLA
fields on 22$^{nd}$ December 2014. The original SLA field for that day interpolated to a spatial
resolution of ⅓° × ⅓° is displayed in the uppermost panel for comparisons purposes. Notice
that the coarse spatial resolution of the 2° × 2° sub-sampled grid (upper-left panel in Figure 5)
prevents us from retrieving the mesoscale features observed in the original map and only the
large-scale signals are properly captured. As a consequence, the RMSE associated with this
reconstruction which simulates the present Argo array in the Mediterranean are around 4.6
cm. On the contrary, the sub-sampled grids with box-sizes of 0.4° × 0.4° and lower (map not
shown) are able to retrieve most of the mesoscale structures of the basin with an RMSE of
around 2.6 cm. Nonetheless, the high number of stations required to reconstruct the SLA maps
(respectively 1458 and 1915, see Table 2) makes this option unviable. Therefore, it is
imperative to reach a compromise between the stations used and the extent of the
reconstruction performed. In this case, a reasonable solution would be to reconstruct the SLA
field from a sub-sampled grid with a box-size of 0.75° x 0.75°. This spatial resolution agrees



with the theoretical one for the Argo array in the Mediterranean extracted from the internal
Rossby radius of deformation computed for the Mediterranean basin. Also, it allows us to
retrieve the most representative mesoscale patterns of the basin, for spatial scales larger than
150 km, with a feasible number of Argo floats (450 stations). Moreover, the spatial scales
resolved by this configuration simulate the spatial scales captured by the altimetry.
**4.3 Sensitivity to the irregular sampling**
The experiments conducted above let us recover SLA maps computed from theoretical
regular-gridded configurations of the Argo array in the Mediterranean. In this section we aim
at retrieving altimetry maps from a realistic configuration of the Argo network by using the
actual uneven positions of the Argo floats in the basin. Figure 6.a displays the real positions of
the 58 Argo floats operating in the Mediterranean Sea on 22$^{nd}$ December 2014. SLA at each
single Argo float position was extracted from the original altimetry map of that day (Figure not
shown). Then, the SLA field for the whole basin was retrieved by following the procedure
applied to the regular-gridded sub-sampled fields.
On the other hand, and since the mean number of Argo floats in the Mediterranean is set
to around 80, random virtual floats were added to the actual Argo array of that day. The aim
was to reach the mean number of platforms normally operating on the basin. The virtual floats
were added by using a normal distribution function computed from the mean and standard
deviation of the positions of the Argo Array in the Mediterranean. Then, the SLA data was
obtained at the locations of both the actual and virtual floats (see Figure 6.b). We kept on
adding random virtual floats until reaching an Argo array of 150, 250 and 450 stations. Their
locations and the corresponding SLA data extracted at each position are respectively displayed
in figures 6.c, d and e. SLA field for the whole basin was then recovered for each configuration
of the Argo array according to the procedure described in the section 4.1. Reconstructed SLA
fields were compared with the original altimetry map of that day. Figure 7 summarizes the
results obtained from both the uneven and regular-gridded experiments conducted on 22$^{nd}$
December 2014. The errors associated with the SLA maps recovered from the different
configurations of the Argo array (gray triangles) present a maximum RMSE of nearly 5 cm
when only the 58 Argo floats operating that day are used to reconstruct the SLA field. As
expected, RMSEs decay as the number of Argo floats increases (notice that here an Argo array
configuration with 750 floats has been also included in order to have a better overview of their
general pattern). This decrease follows the same pattern that the RMSEs obtained from the




regular-gridded experiments (black line) although larger values are observed here. This fact is
related to the uneven spatial distribution of the Argo platforms in the basin.
**5- Discussion**
The Argo network in the Mediterranean Sea consists presently of around 80 operating
floats drifting with less than 2 degrees mean spacing. Even though this array improves the
global coverage of the Argo network, it only captures the large-scale circulation features of the
basin. In this work, we have investigated which configuration in terms of the spatial sampling
of the Argo array in the Mediterranean would be necessary to recover the mesoscale dynamics
in the basin as seen by altimetry. The monitoring of the mesoscale features is not an Argo
program target. However, this issue is of concern since it can help the current ocean state
estimates.
To do that, we have conducted several Observing System Simulated Experiments (OSSEs) in
the basin. The errors of Argo – SLA differences required to perform the OSSEs were obtained
through the comparison of SLAs from altimetry and DHAs computed from the in-situ Argo
network. The comparisons have been focused on the sensitivity to the reference level (400
dbar or 900 dbar) used in the computation of the Argo dynamic height. We found that the
number of Argo profiles reaching 900 m used to compute DHA is almost 6 times smaller than
those reaching 400 m. Therefore, the choice of the reference depth has repercussion in the
number of valid Argo profiles and thus in their temporal sampling and the coverage of the Argo
network used to compare with altimeter data. In addition, the computation of the differences
between altimetry and Argo data referred to both 400 and 900 dbar revealed a standard
deviation of SLA – DHA differences 1.67 cm lower (in terms of variance) when computing DHA
referred to 400 dbar. This fact, together with both a higher correlation coefficient between
both datasets and the larger number of available profiles, make that 400 dbar should be
considered as reference level to compute DHA from Argo data in the Mediterranean basin.
This leads to a standard deviation of the differences between both datasets of 4.92 cm
(equivalent to 90 % of SLA signal variance). Conversely, one would expect better results when
using 900 dbar as reference level because the physical content (variance) of a larger fraction of
the water column is considered when computing Argo DH. However, the climatology used here
to compute DHA could be not as accurate at that level due the lower number of historical data
available at that depth, then resulting in larger standard deviations of the differences between
both datasets.





Another interpretation of the results obtained here could be done in terms of the
dynamics of the water masses residing in the Mediterranean Sea. Due to the excess of
evaporation over precipitation and river run-off, an Atlantic inflow through the Strait of
Gibraltar is required to balance the salt and freshwater budgets of the basin. As the Atlantic
water spreads into the Mediterranean, it becomes saltier and denser under the influence of
intense air-sea interactions [*Criado-Aldeanueva et al.*, 2012]. Most of this flow will return to
the Atlantic Ocean as Levantine Intermediate Water (LIW), formed during winter convection in
the Levantine sub-basin while another part will be transformed into deep waters along the
basin [*Criado-Aldeanueva et al.*, 2012]. The LIW spreads over different fractions of the water
column along its path towards the Atlantic Ocean. In some regions of the Mediterranean the
reference level of 400 dbar (near 400 m depth) would be close to the interface between this
water mass and those residing at deeper levels, which usually have different pathways. As a
consequence, velocities around 400 m depth would be significantly reduced as a result of
friction while they could be enhanced as we move towards deeper levels fed by the
Mediterranean deep water masses. As a result, velocities at 900 m depth could not be close to
zero, as we assume in the DHA computation, then promoting coarser results when comparing
altimetry with Argo data referred to 900 dbar. The depth of the LIW core in most of the
Mediterranean basin is also the reason of choosing 350 m as the parking depth for the Argo
floats in the Mediterranean [*Poulain et al.*, 2007].
Results reported from the regular-gridded experiments have shown that the reconstructed
SLA maps from a configuration similar to the current Argo array in the Mediterranean (spatial
resolution of 2° × 2°) are not able to capture the mesoscale features of the basin. As a
consequence, these maps only retrieve 21 % of SLA signal variance. This is an expected result
because the initial target of the Argo program is to monitor the large-scale ocean variability.
Quite the opposite, reconstructed SLA fields from a 0.75° x 0.75° grid box of SLA observations
retrieve 66 % of SLA signal variance. This reconstruction captures the large-scale signal and
most of the mesoscale features of SLA fields in the basin exhibiting a mean RMSE lower than 3
cm (equivalent to 34 % of SLA signal variance). In addition, this spatial resolution agrees with
the theoretical one extracted from the internal Rossby radius of deformation computed for the
Mediterranean basin. The same outcomes were also obtained from the experiments
conducted by using the actual positions of the Argo array in the basin. Here, larger values for
the RMSEs of the recovered SLA maps were systematically obtained due to the uneven spatial
distribution of the Argo platforms in the basin. However, we must be cautious about these
results because the test has been conducted only along one Argo cycle (10 days). Anyway,





similar results to the ones obtained here are expected to emerge from longer experiments
according to the outcomes obtained from the analysis of 2014 yearly RMSEs associated with
the altimetry maps recovered from the different regular-gridded sub-sampled fields.

4       To summarize, and in light of a hypothetical future expansion of the Argo mission towards

an increase in the spatial sampling resolution, the actual Argo array in the Mediterranean Sea
might be enlarged until reach a spatial resolution of nearly 75 × 75 km according to the results
of the simulation experiments. Such Argo array, equivalent to around 450 floats, cycling every
10 days would be enough to retrieve the SLA field with an RMSE of 3 cm for spatial scales
higher than 150 km, similar to those captured by the altimetry. This array would also have a
net impact on numerical models that assimilate Argo profiles.

## 11   **Acknowledgements**

The research leading these results has received funding from the European FP7 under the
E-AIMS (Euro-Argo Improvements for the GMES Marine Service) project (Code: 312642) and
the Sea Level Thematic Assembly Center (SL-TAC) of the Copernicus Marine and Environment
Monitoring Service (CMEMS). Argo data are collected and made freely available by the
International Argo Program and the national programs that contribute to it
(http://aego.ucds.edu and www.jcommops.org/argo). Altimetry data are generated, processed
and freely distributed by CMEMS (http://marine.copernicus.eu/).

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

gauges and Argo Profiling Floats for data quality assessment and Mean Sea Level studies, *Marine
Geodesy* Vol. 35 Suppl. 1.





| | All valid profiles (DHA ref. 900 dbar) | | Profiles reaching 900m (DHA ref. 400 dbar) | | All valid profiles (DHA ref. 400 dbar) | |
|---|---|---|---|---|---|---|
| **Argo Floats** | 23 | | 24 | | 41 | |
| **Argo Profiles** | 416 | | 479 | | 2258 | |
| **std (SLA-DHA,cm)** | 5.31 | 0.20 | 5.04 | 0.17 | 4.92 | 0.07 |
| **R (SLA-DHA)** | 0.80 | 0.02 | 0.82 | 0.02 | 0.76 | 0.01 |

Table 1: Comparison of correlation and standard deviation (cm) of the differences between new
AVISO product for the Mediterranean Sea and Argo data referred to both 400 dbar and 900 dbar
(sub-columns on the left). Sub-columns on the right display the results of the robustness
experiments in terms of standard deviations (see text for details). DHA referred to 400 dbar has
been computed for the whole valid Argo profiles and those reaching 900 m depth for
comparison purposes. The number of Argo platforms and vertical profiles used are also showed.



| Spatial resolution (degrees) | Number of stations | Filtering scale (km) |
|---|---|---|
| 2°×2° | 69 | 445 |
| 1.5°×1.5° | 121 | 333 |
| 1°×1° | 273 | 225 |
| 0.75°×0.75° | 482 | 167 |
| 0.5°×0.5° | 1082 | 111 |
| 0.4°×0.4° | 1458 | 95 |
| 0.3°×0.3° | 1915 | 82 |
| **0.125°×0.125°** | **17283** | — |

Table 2: Spatial resolution (degrees) and associated number of stations of the different sub-
sampled fields used to reconstruct the SLA in the Mediterranean. The lower line displays the
spatial resolution and stations of the original altimetry maps. The filtering scale (km) used to
compute the recovered SLA fields in the different reconstructions have been also included.





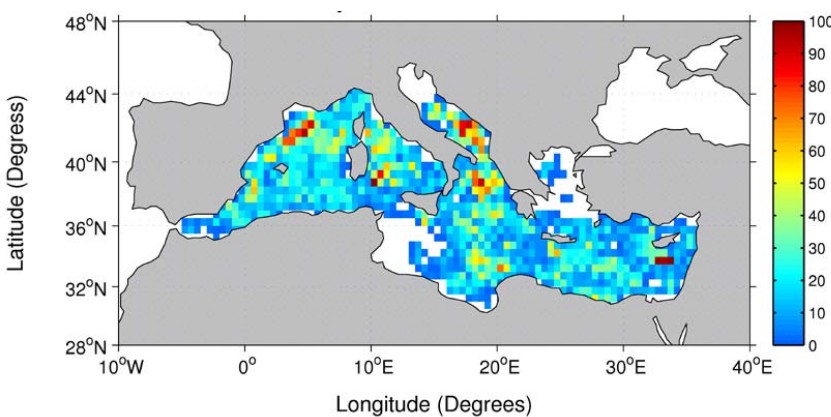

2    Figure 1. Number of Argo profiles on boxes of 0.5°× 0.5° of lat-lon performed between 2003

3    and 2015 in the Mediterranean Sea and used to compute Argo DHs. Only profiles with a

4    position quality flag of 1 (good data) have been considered.





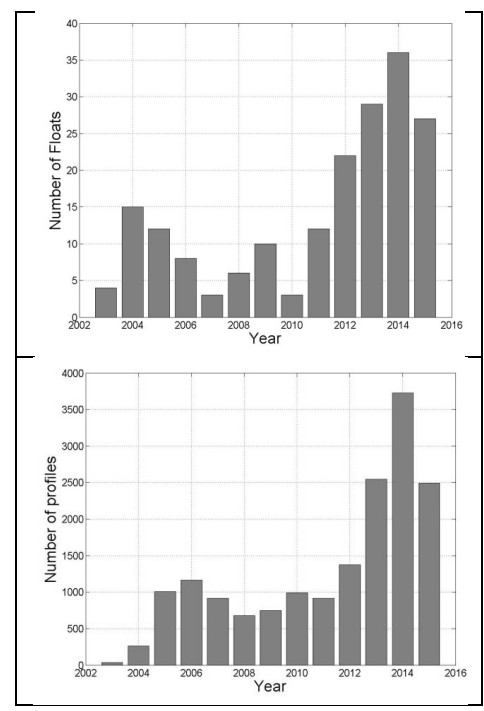

2 Figure 2: Temporal evolution of Argo floats (upper panel) and Argo profiles (lower panel) with

3 a position quality flag of 1 deployed in the Mediterranean Sea since 2003 until the middle of

4 2015.





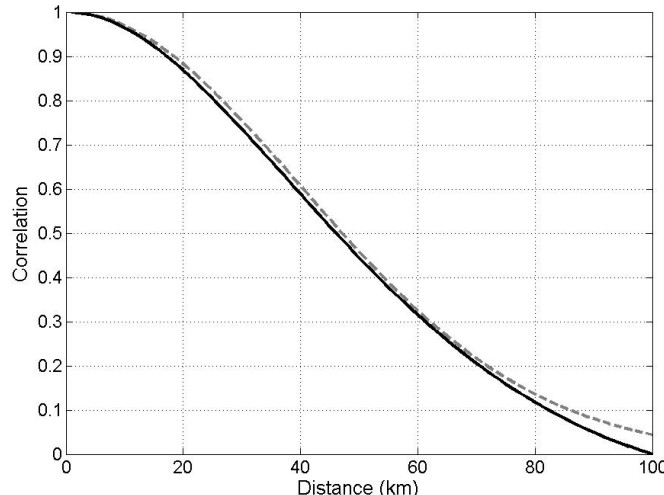

Figure 3: Correlation curve computed for altimetric data (black solid line) for a typical zonal
scale of correlation for the Mediterranean region of 100 km. The gray dashed line shows the
best fitting correlation curve obtained for the reconstruction experiments. It corresponds to a
spatial scale of correlation of 40 km.





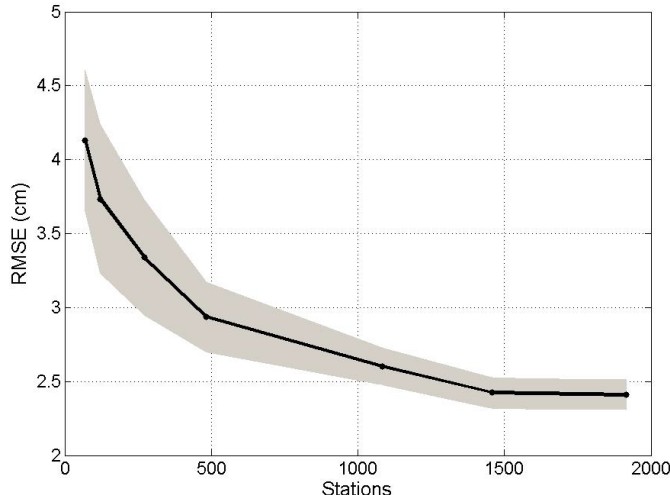

2 Figure 4: Root mean square errors (cm) associated with the altimetry maps recovered along

3 2014 from the different regular sub-sampled fields mentioned in the text. The black line

4 represents the yearly mean value and the gray patch stands for the annual variability.



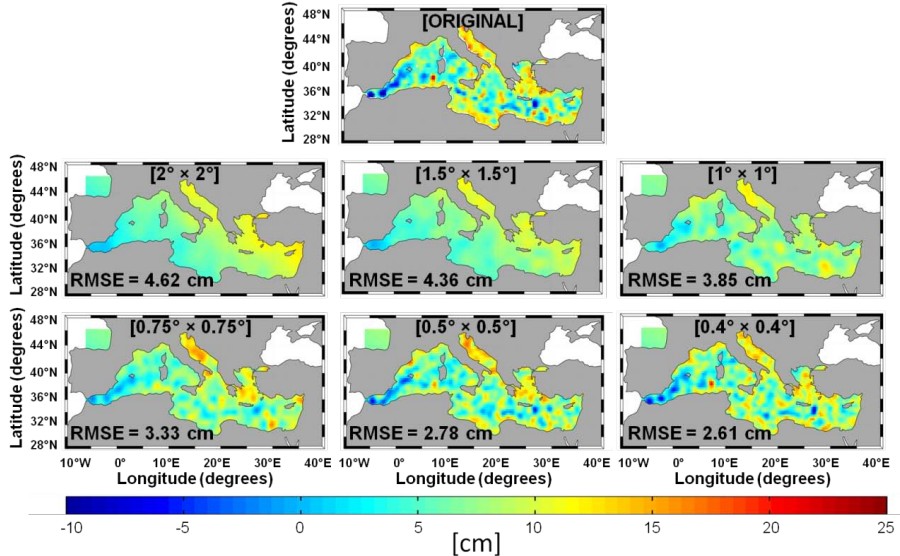

Figure 5: Altimetry maps recovered from the different sub-sampled SLA fields (cm) on December 22, 2014. The spatial resolution of the different regular grids and the RMSEs associated with each reconstruction for that day are also indicated. Moreover, the original SLA field of that day interpolated to a spatial resolution of ⅓° × ⅓° is displayed in the uppermost panel for comparison purposes.



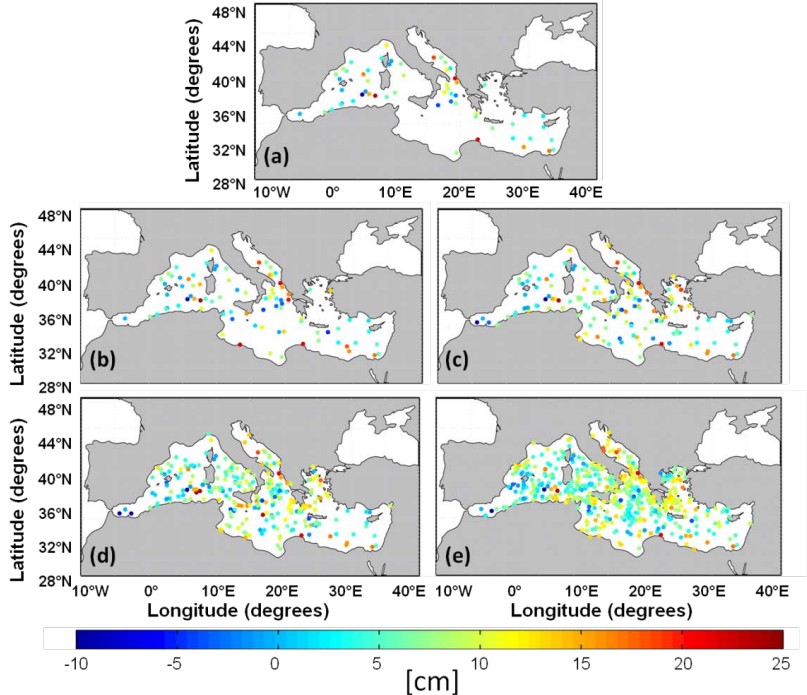

Figure 6: (a) actual positions of the Argo array operating in the Mediterranean basin on December 22, 2014 (58 floats). Colors indicate the SLA (cm) extracted at those locations from the original altimetry map of that day. Panels (b), (c), (d), and (e) display the original Argo array enlarged with random virtual floats in order to simulate an Argo array configuration of 84, 150, 250 and 450 floats, respectively.





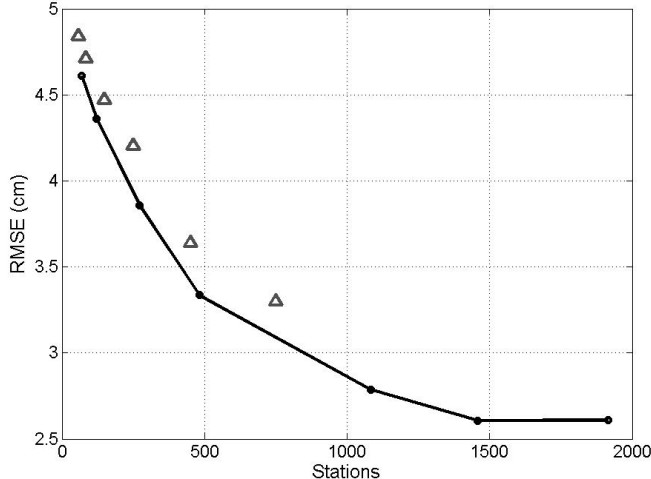

Figure 7: Root mean square errors (cm) associated with the altimetry maps recovered on
December 22, 2014 from the different regular sub-sampled fields mentioned in the text (black
line). Triangles stand for the errors associated with the SLA fields retrieved for that day from
the different configurations of the Argo array in the Mediterranean Sea (see Figure 6). Notice
that an Argo array configuration with 750 floats has been also included for comparison
purposes.

