# Peer review of "On the mesoscale monitoring capability of Argo floats in the Mediterranean Sea"

_Ocean Science, 2016_

## Referee Comment (RC1) · Anonymous Referee #1 · 20 Oct 2016

**1   Overall assessment**

The paper entitled "On the mesoscale monitoring capability of Argo floats in the Mediterranean Sea" by Sanchez-Roman et al presents a topic relevant to the journal. This work would suggest the optimal ARGO network coverage needed to sample the mesoscale activity of the Mediterranean Sea. It must be pointed out that in the framework of E-aims project this aspect has been already addressed. All the participant members agreed that a continuous increase of ARGO coverage in term of horizontal/vertical and temporal resolution is desirable but not easily maintained. In this work the authors used Observing System Simulation Experiment (OSSE) (Arnold and Day, 1986; Kindle, 1986) a well-known procedure oriented to evaluate the impacts of input parameters and sampling scheme in an existing observing system. Observations for

these experiments are simulated so they represent ***optimistic real data*** (Griffa at al, 2007). In this way can be assessed the impact of synthetic data since truth state is known in advance (and everywhere needed). This pioneering investigation through OSSE could give information on the best floats coverage and an estimate of the ***ideal maximum error reduction***. Argo profiling depth and cycles length are crucial parameters in optimizing the energy storage of the floats. A detailed study performed trough OSSE, varying those quoted parameters, was lead by P.-M. Poulain and M. Solari in 2009, and focused to monitor the thermoaline variability in Mediterranean Sea. An estimate of the optimal time cycling for MEDARGO floats was defined by C.Pizzigalli and V.Rupolo (2007). In this work they maximized independent observations and minimize the velocity error at the parking depth. They also analyzed the interannual variability of Lagrangian transport in Western Mediterranean from 2000 to 2004. In this paper the final 3cm RMSE value obtained with 75km box side can be considered as the maximum skill that could be achieved increasing the Argo coverage. For this reason I think that the authors should stress that 75km box side would increase the actual network cost of 6 times in order to have a theoretical maximum reduction of the 40% of the actual RMSE. The scientific methods are not clearly outlined and assumptions are not always valid. Traceability of results is very difficult. Authors give proper credit to referred work but not clearly indicate their own contribution. The overall presentation is not clear. The mathematical background is not explained and not well referred (see also specific comments). There is a lack of supplementary data. I also recommend a flow chart for the methodology developed, since this is the main point of the paper. Which data are employed in which point of the methodology. Looking forward for an improved version of the paper.

[Figure]

**2   Specific comments**

The paper describes in section 2.1 the ARGO dataset, one would ask why "only pro-files with a quality position flag of 1 were employed. The major restriction, however, comes from the salinity data close to the sea surface. Profiles exhibiting salinity flags of 3-4 in the first 5 meters of the water column were removed before DH computation". The quality flag for position is for sure a good way to remove gross bad data, but a second quality check should be done on pressure, temperature and salinity, if one of the corresponding flags is different from 1 the data both T/s at that pressure should be neglected (not only salinity in the first 5 meters). Do the authors check the stability of the ARGO profile? Do the authors check the spikes? In our operational system more than the 20% of input data in NRT are neglected for quality check procedure. In section 2.2 the authors describes the altimeter measurements used referring to Pujol et al 2013 and Pujol et al 2016. Here the time period spans from 1993 to 2014 while in the previous section it seems that ARGO are available from 2003 to 2015. And than the authors do a sort of time homogenization removing a mean altimetric map from 2003 to 2011. I think that some more attention should be paid in this operation or at least it should be better explained. Moreover the heuristic value of 3.54cm coming from data should be well described. In section 3 there isn't a clear description, reference or formula that authors used to evaluate the steric effect from Argo float. Moreover at the beginning it seems that "deeper the reference level (of null velocity), the more infor-mation form T/S profiles is consider" but the standard deviation evaluated at 400dbar and 900dbar contradict this sentence (the correlation doesn't, instead). This result is discussed in section 5, but proofs aren't provided. For example if the problem is the different time reference of the dataset a study should be done in order to have them referred to the same value. Conversely in the second hypothesis the authors suggest that velocity at 400dbar or 900dbar are not zero as expected by theory. Why don't they evaluate this term from an operational ocean model? Section 4.1 that should describe the experiment design is a mess. In OSSEs there are some basic step that should be

well described:

- "TRUTH" - the Nature Run or observation;

- What PERTURBATION is used to simulate our incomplete knowledge of the Sea;

- How "synthetic observations" come from the Nature fields

- The synthetic observations are then used in the Perturbation quantify improvements or whatever the aim is.

I think that truth is SLA map at 1/8 resolution. The authors add a random value that corresponds to the sum of instrumental and representativeness error (proof?).

1. The instrumental error for satellite is well defined and is around 2-3cm and depends on satellite (jason1-2 are better for med sea compared for example with cryosat or altika) it can be provided by CLS;

2. Representativeness can be evaluated as Oke and Sakov 2008, re-gridding SLA map at different resolution.

A good way to proceed would be to evaluate the sum of instrumental error and representativeness, if this value is in agreement with the 4.92cm evaluated from author the OSSE can represent a good test bed, otherwise more careful should be paid. Moreover there are some areas in the Mediterranean Sea more dynamic than the others, there I doubt that OSSE is a good approximation. Finally, the authors describe different time window for ARGO and SLA but the OSSE is referred to the whole 2014. So the authors use a perturbation evaluated in a period and apply it in another period. Is it right? According to me this is not a good scientific way to proceed.

**3 Other:**

- Numbers should be avoid in the abstract

- Page 2, lines 30-31: The Sicily channel separates the eastern and western basins [Criado- Aldeanueva et al., 2012] This sentence doesn't have meaning. The authors should say the Sicily strait being only 300-400m meter deep divide Med sea in 2 sub-basin circulation patterns. The western basin is influenced by Gibraltar inflow, the eastern is driven by winds and the consequently LIW formation.

- Page 3, line 17:" Argo and satellite altimetry are entirely complementary" This is not false, but not fully truth. They are different type of measurement in situ and remote sensing and sample different aspect and quantity of Mediterranean Sea.

---

## Referee Comment (RC2) · Anonymous Referee #2 · 16 Dec 2016

The authors present an analysis of Argo float deployment strategies in the Mediterranean Sea. They generate synthetic Argo profiles using satellite altimetry maps based on statistical parameters derived by comparing altimetry maps to actual Argo profiles. Field reconstruction techniques are then used to produce horizontal altimetry maps from synthetic profiles sampled at different horizontal resolutions. Errors between original and reconstructed maps are then calculated to assess the impact of reducing the horizontal spacing between Argo floats.

There is one major issue that the authors need to consider before the paper is published. The analysis procedure described in this paper is not a rigorous OSSE. Rigorous OSSE procedures have been developed in the meteorology community and are only recently being transitioned to the ocean. A recent paper of interest by Hoffman and Atlas (2016) that describes rigorous OSSE procedures can be found at the following
link:

http://journals.ametsoc.org/doi/full/10.1175/BAMS-D-15-00200.1

Click on the 'full text' tab for the main paper and on the 'supplementary materials' tab for the checklist for validating an OSSE system and executing rigorous OSSEs.

These comprehensive OSSE procedures have been developed to insure that the resulting impact assessments are credible and unbiased. A key step toward validating an OSSE system is given in the Hoffman and Atlas paper: "An important component of the OSSE that improves the interpretation of results is validation against a corresponding OSE. In this regard, the accuracy of analyses and forecasts and the impact of already existing observing systems in simulations is compared with the corresponding accuracies and data impacts in the real world. This ensures that the results of the OSSEs are credible and realistic." A first example of applying this validation step in the ocean is presented in the Halliwell et al. (2014) reference contained in the Hoffman and Atlas paper.

The results contained in the present paper are interesting and should be published. It is too much to expect that the authors develop and validate a comprehensive OSSE system at this time. However, these results should be placed in context with regard to state-of-the-art OSSE systems that enable rigorous validation of results. Such rigorous validation is not possible with the approach used in this paper, which perhaps should be referred to as a "simplified OSSE approach". It therefore should be made clear that these results represent a first look that needs to be validated in the future with a comprehensive OSSE system.

---

## Author Comment (AC1) · 13 Jan 2017

Firstly, we wish to thank the reviewer for providing interesting and constructive comments to this paper.

Detailed response to comments of reviewer 1:

Overall assessment:

Reviewer comment: In this paper the final 3cm RMSE value obtained with 75km box side can be considered as the maximum skill that could be achieved increasing the Argo coverage. For this reason I think that the authors should stress that 75km box side would increase the actual network cost of 6 times in order to have a theoretical maximum reduction of the 40% of the actual RMSE. The scientific methods are not

clearly outlined and assumptions are not always valid. Traceability of results is very difficult. Authors give proper credit to referred work but not clearly indicate their own contribution. The overall presentation is not clear. The mathematical background is not explained and not well referred (see also specific comments). There is a lack of supplementary data. I also recommend a flow chart for the methodology developed, since this is the main point of the paper. Which data are employed in which point of the methodology.

Response: In the new version of the manuscript we have emphasized in the abstract and the discussion section that the proposed enhancement of the spatial coverage of the Argo network in the Mediterranean Sea to reach a RMSE of 3 cm would also promote an increase of the actual network cost according to the suggestion of the referee. We have re-written the methodology used to conduct the OSSEs in order to clarify and avoid confusion (see response to specific comments). Also, a flow chart showing the methodology has been added (figure 3 in the new version). We have included the following sentence in the first paragraph of the experiment design section: "This section describes the different elements of the OSSEs conducted in the Mediterranean Sea. A flow chart of the methodology developed is provided in Figure 3". Moreover, we have included two tables as supplementary data to support the choice of the reference level for the DHA computation in the Mediterranean Sea based on the dynamics of the water masses residing in the basin.

Specific comments:

Reviewer comment: The paper describes in section 2.1 the ARGO dataset, one would ask why "only profiles with a quality position flag of 1 were employed. The major restriction, however, comes from the salinity data close to the sea surface. Profiles exhibiting salinity flags of 3-4 in the first 5 meters of the water column were removed before DH computation". The quality flag for position is for sure a good way to remove gross bad data, but a second quality check should be done on pressure, temperature and salinity, if one of the corresponding flags is different from 1 the data both T/s at that pressure

should be neglected (not only salinity in the first 5 meters). Do the authors check the stability of the ARGO profile? Do the authors check the spikes?

Response: we have used in this study delayed mode quality-controlled T/S profiles as obtained from the Coriolis Global Data Assembly Centre (see section 2.1, lines 17-19 in page 5). Therefore, the stability of the Argo profiles and spikes have been checked previous to their use in this study. However, an additional quality control criterion was applied here before the DH computation in order to remove spurious data. We applied a quality check based on (i) the profile position and (ii) the pressure, temperature and salinity flags. Nonetheless, we found flags different from 1 only in the uppermost part of the water column and specifically for the salinity data. We realize that the sentence mentioned by the reviewer is unclear in the paper and it has been re-worded in the new version of the manuscript to avoid confusion as follows:

"An additional quality control criterion relative to both the profile's position and the pressure, temperature and salinity measurements was applied: only profiles with a quality position flag of 1 (good data) were employed. Moreover, data exhibiting temperature and/or salinity flags different from 1 were removed before the DH computation."

Reviewer comment: in section 2.2 the authors describe the altimeter measurements used referring to Pujol et al 2013 and Pujol et al 2016. Here the time period spans from 1993 to 2014 while in the previous section it seems that ARGO are available from 2003 to 2015. And then the authors do a sort of time homogenization removing a mean altimetric map from 2003 to 2011. I think that some more attention should be paid in this operation or at least it should be better explained. Moreover the heuristic value of 3.54cm coming from data should be well described.

Response: The datasets used in this paper span over different time-periods: Argo data is available from 2003 to 2015 while Altimetry data span the period 1993 to 2014. On the other hand, the climatology used to compute DHA from Argo data is available from 2003 to 2011. We must choose a common period for the Argo and Altimetry datasets

before the comparisons. In this case, we select the time period spanning between January 2003 (beginning of the Argo dataset) and April 2014 (ending of the altimetric data). Moreover, to conduct the comparisons, both altimetry and Argo data must have the same interannual temporal reference. The temporal reference of Argo data is given by the climatology that we used to compute DHA (from January 2003 to December 2011). Therefore, to have the same temporal reference in altimetry, we must subtract the mean SLA over 2003 – 2011 from the original altimetric maps. Furthermore, in the new version of the manuscript we have removed the heuristic value corresponding to the mean SLA for the Mediterranean basin over the subtracting period (2003-2011) since this value is not relevant for the computation. We subtract the mean value of SLA over this period at each single node of the grid and not the mean SLA of the whole basin. The paragraph has been re-worded in the new version as follows in order to clarify:

"Notice that the availability of altimetry and Argo data does not match. Therefore, a common period spanning the period from January 2003 (beginning of the Argo dataset) to April 2014 (ending of the altimetric data analysed in this study) has been used in both datasets. Moreover, to perform this comparison, it is critical that Altimetry and Argo data have the same interannual temporal reference [Legeais et al., 2016]. We estimate DHAs from Argo data through a synthetic mean Argo dynamic height referred to the time period between 2003 and 2011. Thus, the temporal reference of the altimeter SLA must be adapted to this time period. To do that, we subtract the mean of altimetric SSALTO/DUACS maps over 2003 âŤĂ 2011 from the original SLA time series [Valladeau et al., 2012]."

Reviewer comment: In section 3 there isn't a clear description, reference or formula that authors used to evaluate the steric effect from Argo float.

Response: the computation of DHA from Argo data referred to a given pressure level comprises the steric effect corresponding to the fraction of the water column considered. Therefore, the deeper the reference level, the better sampled the steric signal is

through the water column. A further explanation is given in the next comment.

Reviewer comment: "deeper the reference level (of null velocity), the more informa-tion from T/S profiles is consider" but the standard deviation evaluated at 400dbar and 900dbar contradict this sentence (the correlation doesn't, instead). This result is dis-cussed in section 5, but proofs aren't provided. Conversely in the second hypothesis the authors suggest that velocity at 400dbar or 900dbar are not zero as expected by theory. Why don't they evaluate this term from an operational ocean model?

Response: the choice of a deep reference level for Argo DHAs provides a better es-timation of the baroclinic signal and it is more in agreement with the observed signal by altimetry. Therefore, lower values of STD of the differences between SLA from al-timetry and DHA from Argo data referred to 900 dbar are expected. Nonetheless, we found that DHA referred to 400 dbar promote lower values of STD diff. even though the vertical steric signal is less sampled. We suggest that the larger coverage of the Argo network when computing DHA referred to 400 dbar due to the more comprehen-sive number of available Argo profiles (the number of Argo profiles reaching 400 m is almost 6 times larger than those reaching 900 m) plays a more crucial role in the com-parisons with altimetry in the Mediterranean Sea than the deep sampling of the steric signal. We have added the following sentence in section 5 of the new version of the manuscript:

"Conversely, one would expect better results when using 900 dbar as reference level because the physical content (variance) of a larger fraction of the water column is considered when computing Argo DH. However, the more comprehensive number of available Argo profiles when using 400 dbar as reference level, and thus the larger coverage of the Argo network, seems to play a more critical role in the comparisons with altimeter data in the Mediterranean basin than the deep sampling of the steric signal. On the other hand, the climatology used here to compute DHA could be not as accurate at 900 m due the lower number of historical data available at that depth, then resulting in larger standard deviations of the differences between both datasets."

The second part of the reviewer's comment is related to the second hypothesis proposed here to explain our results in terms of the dynamics of the water masses in the Mediterranean Sea. Zavatarelli and Mellor [1995] describe the spreading of LIW in the water column throughout the Mediterranean basin along its path towards the Atlantic Ocean. These authors stated that this water mass is located between 100 – 400 m depth in the eastern basin while it spreads between 200 – 700 m approximately in the western basin. We have added the following sentence in the new version of the manuscript to both clarify the distribution of this water mass along the Mediterranean basin and support our subsequent interpretation of the outcomes obtained here:

"The LIW spreads over different fractions of the water column along its path towards the Atlantic Ocean: in the eastern basin it is located between 100 – 400 m depth while it spreads between 200 – 700 m approximately in the western basin [Zavatarelli and Mellor, 1995]."

Furthermore, to check this hypothesis we recomputed the STD diff. (SLA - DHA) for the eastern and western basins and we compared these outcomes with the vertical distribution of LIW in the Mediterranean. We found lower values of STD diff. in the two sub-basins when computing DHA referred to the reference level close to the lower bound of the LIW (400 dbar in the eastern basin and 900 dbar in the western Mediterranean) where velocities close to zero are expected. This outcome supports the hypothesis stated here. We have included the corresponding tables in the new version of the manuscript as supplementary material and the following paragraph has been added to section 5:

"In order to check this hypothesis, we recomputed the SLA - DHA differences for the eastern and western basins (see Tables S1 and S2 in the supplementary data). In a first step, the Argo profiles available to compute DH in the whole Mediterranean were sorted out according to their location. We found that 44 % of them were deployed in the western Mediterranean while the remaining 56 % are located in the eastern basin. Then, DHA referred to 400 and 900 dbar was computed and compared with SLA from

Altimetry according to the procedure described in section 3. In the eastern Mediterranean, the computation of the differences between altimetry and Argo data referred to both 400 and 900 dbar revealed a standard deviation of SLA - DHA differences 1.88 cm lower (in terms of variance) when computing DHA referred to 400 dbar. This pressure level is located nearby the bounds of the LIW in this region, where velocities close to zero are expected. By contrast, in the western basin we obtained a standard deviation of SLA - DHA differences 1.26 cm lower when computing DHA referred to 900 dbar. This result is consistent with the vertical distribution of the LIW in the western Mediterranean."

Reviewer comment: Section 4.1 that should describe the experiment design is a mess.

Response: as it was aforementioned in the response to the overall assessment, the section "experiment design" has been re-written in the new version of the manuscript to better describe the basic steps followed to conduct the OSSEs in the Mediterranean Sea. We have included the description of the Nature run used in the experiments and how the synthetic observations are obtained from this "truth". Moreover, a flow chart of the methodology developed has been included. We have added the following sentence:

"The specific altimetry gridded merged product for the Mediterranean Sea, described in section 2.2, has been used as the Nature Run (NR) component of the OSSEs. Namely, we use daily SLA maps along 2014. The region considered covers the entire Mediterranean basin. The original altimetry dataset has a spatial resolution of $1/8° \times 1/8°$ and presents 17283 grid points (see Table 2). We obtain synthetic observations from the Nature fields by sub-sampling the NR with the different spatial resolutions displayed in Table 2. The aim is to reproduce some possible configurations of the Argo array network in the Mediterranean Sea."

Details about the perturbation applied to the synthetic observations are provided in the response to the next reviewer comment.

Reviewer comment: The authors add a random value that corresponds to the sum of instrumental and representativeness error (proof?). A good way to proceed would be to evaluate the sum of instrumental error and representativeness, if this value is in agreement with the 4.92cm evaluated from author the OSSE can represent a good test bed, otherwise more careful should be paid.

Response: In the OSSE approach, observation errors have to be added to the observation values extracted from the true field. Section 3 provides a detailed characterization of the differences between altimeter SLA and real Argo DHA, which can be considered as "observation errors" in our particular OSSE experiment where Argo DHA are the observations and altimeter SLA is the true field. Since the differences between altimeter SLA and real Argo DHA are affected by both instrumental errors in Argo profiles and representation errors due to different spatio-temporal resolution of these observations also containing the signature of different physical processes, we consider them as the total error including contributions from both instrumental and representation errors. These ideas have been clarified in the new version of the manuscript. We have added the following sentence:

"In addition, the synthetic observations (re-gridded daily SLA maps) were perturbed simulating realistic observation errors. The differences between altimeter SLA and real Argo DHA directly provide the observation errors in our particular OSSE experiment where Argo DHA are the observations and altimeter SLA is the true field. A random noise generated from a normal distribution function representing the errors characterized in Section 3 but limited to the year 2014 is added to the values of the synthetic observations. The STD difference for the year 2014 is 4.79 cm."

Reviewer comment: Finally, the authors describe different time window for ARGO and SLA but the OSSE is referred to the whole 2014. So the authors use a perturbation evaluated in a period and apply it in another period. Is it right? According to me this is not a good scientific way to proceed.

Response: we thank the reviewer for drawing our attention to this point. We have re-computed the STD of the differences (SLA - DHA) only for year 2014 and we obtained a new STD value of 4.79 cm. We have used this value to perform the OSSEs in the Mediterranean basin. Figure 6 (in the new version) has been produced according to the new outcomes. Notice that the re-computed STD value is very close to the one obtained for the whole period analysed (4.92 cm) so results are quite similar to the previous ones. We have included this new STD value (and the corresponding figure) in the new version of the manuscript. We added the sentence of the previous response:

"In addition, the synthetic observations (re-gridded daily SLA maps) were perturbed simulating realistic observation errors. The differences between altimeter SLA and real Argo DHA directly provide the observation errors in our particular OSSE experiment where Argo DHA are the observations and altimeter SLA is the true field. A random noise generated from a normal distribution function representing the errors character-ized in Section 3 but limited to the year 2014 is added to the values of the synthetic observations. The STD difference for the year 2014 is 4.79 cm."

Other:

Reviewer comment: Numbers should be avoid in the abstract

Response: we think that numbers obtained in this paper are relevant outcomes and therefore they should be kept in the abstract.

Reviewer comment: Page 2, lines 30-31: The Sicily channel separates the eastern and western basins [Criado-Aldeanueva et al., 2012]. This sentence doesn't have meaning. The authors should say the Sicily strait being only 300-400m meter deep divide Med. Sea in 2 sub-basin circulation patterns. The western basin is influenced by Gibraltar inflow, the eastern is driven by winds and the consequently LIW formation.

Response: the sentence has been re-worded in the new version as follows: "The Sicily Strait being only 300 – 400 m depth divides the Mediterranean Sea in two sub-basin

circulation patterns: the western basin is influenced by the Gibraltar inflow while the eastern basin is driven by winds and the consequently LIW formation."

Reviewer comment: Page 3, line 17:" Argo and satellite altimetry are entirely complementary" This is not false, but not fully truth. They are different type of measurement in situ and remote sensing and sample different aspect and quantity of Mediterranean Sea.

Response: this sentence has been modified in the new version as follows: "Argo data complement satellite altimetry."

[Figure]

**Fig. 1.** Figure 3: Flow chart showing the elements of the OSSEs conducted for the Mediter-
ranean Sea. Datasets used in each component are also indicated.

---

## Author Comment (AC2) · 13 Jan 2017

Firstly, we wish to thank the reviewer for providing interesting and constructive comments to this paper.

Detailed response to comments of reviewer 2:

Reviewer comment: There is one major issue that the authors need to consider before the paper is published. The analysis procedure described in this paper is not a rigorous OSSE. Rigorous OSSE procedures have been developed in the meteorology community and are only recently being transitioned to the ocean. These comprehensive OSSE procedures have been developed to insure that the resulting impact assessments are credible and unbiased. A key step toward validating an OSSE system is given in Hoffman and Atlas (2016). The results contained in the present paper are interesting and

should be published. It is too much to expect that the authors develop and validate a comprehensive OSSE system at this time. However, these results should be placed in context with regard to state-of-the-art OSSE systems that enable rigorous validation of results. Such rigorous validation is not possible with the approach used in this paper, which perhaps should be referred to as a "simplified OSSE approach". It therefore should be made clear that these results represent a first look that needs to be validated in the future with a comprehensive OSSE system.

Response: we thank the reviewer for drawing our attention to this point and for the suggested references. To develop and validate a comprehensive OSSE system we would need to sub-sample the Argo array in the Mediterranean Sea in order to have the corresponding OSE. Nonetheless, the low number of currently available Argo floats in the basin makes unfeasible a high-resolution study. For this reason, we have decided to define our approach as a "simplified OSSE approach" according to the suggestion of the reviewer. We have included in section 4 the following sentence to make clear that OSSEs conducted here do not follow the comprehensive procedure developed for the atmosphere and that the results reported need to be validated with a comprehensive OSSE system:

"As most of the ocean OSSEs conducted to date, OSSEs performed here do not follow the comprehensive design criteria and validation methodology developed for the atmosphere [Halliwell et al., 2014]. Rigorous OSSE procedure includes the validation against a corresponding OSE to guarantee the reliability of the outcomes of the OSSEs [Hoffmann and Atlas, 2016]. As a consequence, our approach can be qualified as simplified OSSE. Further validation will be needed in the future implementing a comprehensive OSSE system."

The abstract and discussion have been also modified to include the approach followed and the need of a further validation through a comprehensive OSSE system.